# AmbigDocs: Reasoning across Documents on Different Entities under the Same Name

**Yoonsang Lee**$^{\diamond,\heartsuit,*}$ **Xi Ye**$^{\diamond}$**, Eunsol Choi**$^{\diamond}$

The University of Texas at Austin$^{\diamond}$, Seoul National University$^{\heartsuit}$
lysianthus@snu.ac.kr, xiye@cs.utexas.edu, eunsol@utexas.edu

## Abstract

Different entities with the same name can be difficult to distinguish. Handling confusing entity mentions is a crucial skill for language models (LMs). For example, given the question "Where was Michael Jordan educated?" and a set of documents discussing different people named Michael Jordan, can LMs distinguish entity mentions to generate a cohesive answer to the question? To test this ability, we introduce a new benchmark, AmbigDocs. By leveraging Wikipedia's disambiguation pages, we identify a set of documents, belonging to different entities who share an ambiguous name. From these documents, we generate questions containing an ambiguous name and their corresponding sets of answers. Our analysis reveals that current state-of-the-art models often yield ambiguous answers or incorrectly merge information belonging to different entities. We establish an ontology categorizing four types of incomplete answers and automatic evaluation metrics to identify such categories. We lay the foundation for future work on reasoning across multiple documents with ambiguous entities.[1]

## 1 Introduction

Different entities can be referred by the same name, and clustering such confusing entity mentions requires rich reasoning involving background knowledge. While it is rare for different entities who share the same name to occur together within a single document, such co-occurrences are prevalent in multi-document setting. For example, searching the web with the question "When is ACL 2024?" returns a webpage on Austin City Limits Music Festival and another page on Association for Computational Linguistics conference. Given such two webpages, can LMs generate a complete answer correctly distinguishing confusing entities, such as "Austin City Limits Music Festival will be held on Oct 4-6 and 11-13, while the ACL conference will happen on August 11–16."?

Despite rich study in ambiguous question answering (QA) (Min et al., 2020; Stelmakh et al., 2022), which covers such ambiguous entity resolution, few work studied how LMs reason when provided with a confusing document set. The major obstacle has been the lack of annotated confusing document sets.[2] To enable research in this area, we construct a synthetic dataset, AmbigDocs, whose single instance consists of a question asking about an ambiguous entity and a list of gold document-answer pairs for each disambiguated entity (see Table 8 for examples). We leverage Wikipedia disambiguation pages, which provide multiple entities that can be referred to by the same surface name. From each disambiguation page, we generate a question and multiple valid answers, where each answer is grounded in a document and corresponds to a different entity in the same ambiguation page. We generate a total of 36K examples, covering 102K unique entities over all domains.

---

*Work was done at UT Austin.

[1]Our dataset is released at https://ambigdocs.github.io/.

[2]While newer datasets contain evidence documents for a few examples, we identify only 111 instances for ASQA (Stelmakh et al., 2022) dataset and 106 instances for QuoteSum (Schuster et al., 2023) dataset that cover entity disambiguation.

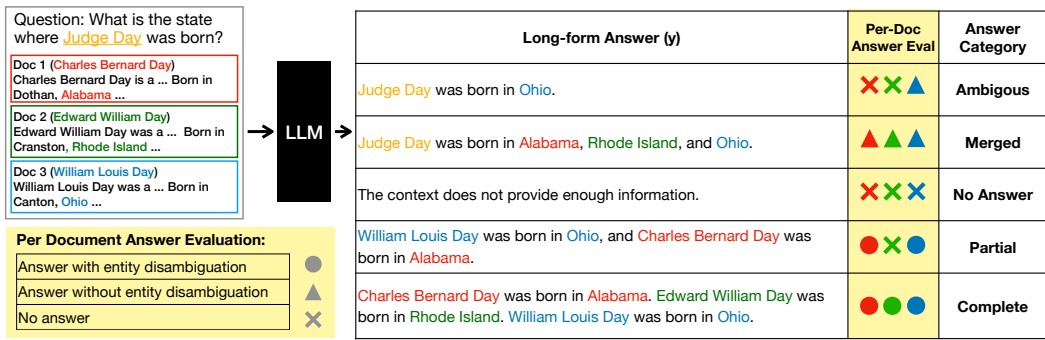

Figure 1: Given a question containing an ambiguous entity mention (e.g., Judge Day) and a set of documents containing a valid answer to different disambiguated entities, LLM should generate a complete answer, pairing each disambiguated entity with its answer. Left bottom box: we first evaluate $y$ with respect to a single input document, checking if $y$ contains the answer with its disambiguated entity name (e.g., for the Doc 1, it should mention the answer "Alabama" and disambiguated entity name "Charles Bernard Day"). Based on per-document scores, we can assign one of the answer category labels (partial, no, ambiguous, merged, complete).

Equipped with a new dataset that contains an ambiguous question and a set of documents with valid disambiguated entity-answer pairs, we evaluate how LMs answer questions containing ambiguous entity names when provided with a set of documents suggesting multiple distinct answers. We first evaluate LMs with metrics used in prior ambiguous QA tasks (Adlakha et al., 2023; Stelmakh et al., 2022). Then, we develop an ontology of five types of answers (complete, partial, ambiguous, merged, and no answer) that captures the unique challenge in AmbigDocs. We develop an automatic metric that can classify answers according to this ontology, with very high agreement of manual classification. We find that both open-source LLMs (Touvron et al., 2023; Jiang et al., 2023) and OpenAI's GPT models (OpenAI, 2023) suffer at identifying different entities, failing to generate complete answers. Yet, we find prompting with in-domain in-context examples can improve model performance significantly, GPT-4/Mistral-7B generating a complete answer for 58%/43% of questions, 20%/10% gain from the zero-shot setting. We provide a rich analysis showing the strengths and weaknesses of existing LMs under this challenging scenario.

To summarize, our contributions are as follows: (1) We present AmbigDocs, a large-scale dataset that pairs a question and a set of documents that suggests multiple answers based on question entity disambiguation. (2) We establish an ontology categorizing five types of generated answers and an automatic classification heuristic, enabling in-depth analysis on model behaviours. (3) We benchmark how current LMs perform the question answering task when provided ambiguous questions and a document set that requires challenging entity disambiguation.

## 2   Overview

**Definition**   We first define key terminology used in the paper.

- **Surface Name ($sn$)**: An ambiguous entity name that can be interpreted as any of disambiguated entities, depending on the context. In Figure 1, $sn$ is "Judge Day".
- **A list of disambiguated entities ($[DE_1, DE_2, \ldots, DE_n]$)**: A set of $n$ distinct entities that share the same surface name $sn$. In Figure 1, this will be [Charles Benard Day, Edward William Day, William Louis Day].
- **Question ($q_{sn}$)**: A question that contains the surface name $sn$. The question is ambiguous as $sn$ can refer to multiple entities, each with different answers. In Figure 1, it is "What is the state where Judge Day was born?".
- **A list of answers ($[a_1, a_2, \ldots, a_n]$)**: Given the question $q_{sn}$, $a_i$ is the answer corresponding to $DE_i$. In Figure 1, it is ["Alabama", "Ohio", "Rhode Island"].
- **A set of documents relevant to the question ($\{d_1, d_2, \ldots d_m\}$)**: A set of $m$ documents that are relevant to question $q_{sn}$ that will be provided to the LMs.

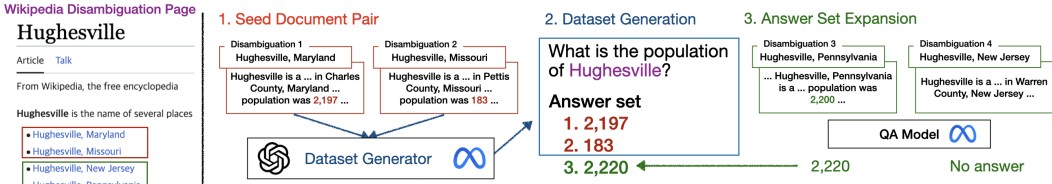

Figure 2: Overview of our dataset generation. We identify a surface name and a list of disambiguated entities from Wikipedia's disambiguation pages. We select two documents for generating a question and their corresponding answers. Subsequently, we gather additional answers from the remaining documents.

**Task** We study retrieval-augmented generation (Lewis et al., 2020; Izacard & Grave, 2020; Sachan et al., 2021), which allows LMs to incorporate external documents at the inference time. Given a question and a set of relevant documents, $\{q_{sn}, \{d_1, ...d_m\}\}$, LMs are tasked with generating a long-form answer, $y$,[3] from which we can infer a set of disambiguated entities $[DE_1, DE_2, \ldots, DE_n]$ and their corresponding answers, $[a_1, a_2, \ldots, a_n]$. The key difference from prior work on ambiguous QA (Min et al., 2020; Stelmakh et al., 2022) is that we provide a set of relevant documents, which contains multiple disambiguated entities and their corresponding answers. This allows an isolated evaluation of LMs on their ability to reason with a confusing document set, without conflating the retriever performance.[4]

**Evaluation** We propose a reference-based evaluation. Given the reference set of disambiguated entities $[DE_1^*, DE_2^*, \ldots, DE_m^*]$ and their corresponding answers, $[a_1^*, a_2^*, \ldots, a_m^*]$ and predicted answer $y$, we categorize the answer $y$ into one of the five categories that we define:

- **Complete Answer**: LM correctly generates all valid answers in the provided document set. Each answer is paired with the disambiguated entity name.
- **Partial Answer**: LM generates one or more (but not all) of the valid answers in the provided document set. Each answer is paired with the disambiguated entity name.
- **No Answer**: LM abstains from providng any answer.
- **Ambiguous Answer**: LM generates one of the valid answers without providing the disambiguated entity name.
- **Merged Answer**: LM generates multiple answers without providing the disambiguated entity name (merging facts about multiple disambiguated entities).

Figure 1 provides an example answer corresponding to each output category. Ideally, LM should generate a complete answer. Merged answer can be considered the most problematic, as it may provide confusing or misleading information. The preference among partial answer, no answer, and ambiguous answer can depend on the types of questions and the prominence among disambiguated entities. If one of the disambiguated entities is substantially more likely to be referred by surface name, providing an ambiguous answer corresponding to the likely entity can be more useful than abstaining. In Section 4.2, we will introduce an automatic metric that assigns model outputs to one of the five answer categories. Now we describe how we construct AmbigDocs to study this problem.

## 3  Creating AmbigDocs

We build an automated data creation pipeline that can generate large-scale data without manual annotations, once provided with Wikipedia disambiguation pages and access to high-quality LLMs. To ensure the quality of the synthetic data, our pipeline features carefully crafted prompts for sampling initial instances and various checks for filtering low

---

[3]Each answer consists of roughly 20 to 60 words (answer length statistics in Table 2).

[4]We additionally provide experimental results with retrieved document sets, but the main analysis is performed with the gold document set.

quality instances. This follows recent work on generating synthetic datasets for complex tasks with LMs (Xie et al., 2023; Yehudai et al., 2024; Oh et al., 2024; Zhao et al., 2024).

**Data Instance** Each instance in our benchmark consists of {a surface name $sn$, a question containing the surface name $q_{sn}$, and a set of gold documents ($\{d_1^*, d_2^*, \ldots d_m^*\}$). Each gold document $d_i^*$ contains a disambiguated entity $DE_i^*$ and its corresponding answer $a_i^*$.

**Overview** We generate data by first identifying two documents, each belonging to a disambiguated entity under the same surface name. Next, we generate a question and two answers; each answer belongs to a disambiguated entity. This diverges from prior work on this domain that applies heuristics to annotate gold documents for existing questions (Stelmakh et al., 2022; Schuster et al., 2023), which leads to incomplete document annotation. Figure 2 summarizes our process. The first step is to identify $(d_1^*, d_2^*)$, two documents belonging to different entities which can be referred to by the same surface name and discuss similar topics (Sec. 3.1). The second step is to generate a question $q_{sn}$ and answers corresponding to respective entities $\{a_1^*, a_2^*\}$ (Sec 3.2). As the second step only considers two disambiguated entities per surface name, we expand the answer set to gather remaining plausible answers (Sec 3.3).

## 3.1 Generating Seed Document Pair

For each surface name $sn$, we have a set of documents $\mathcal{D}$ (10.6 documents on average) gathered from Wikipedia disambiguation page, each belonging to one of the disambiguated entities. We use the Wikipedia snapshot from December 20th, 2018, where each article are split into 100 words (which we consider as documents) (Karpukhin et al., 2020). For each disambiguated entity of the snapshot, we collect the documents where the disambiguated entity serves as the title. Our goal is to find two documents discussing a similar topic, belonging to different disambiguated entities, each containing a distinct answer to some ambiguous question $q_{sn}$.

For each document pair, we measure their longest $n$-gram overlap as $n$. We define a **relevancy score** between a pair of documents based on $n$. The relevancy between a pair is set to 0 if $n \leq 3$ or $n \geq 10$; we set the relevance as $n$ otherwise. Here we set relevancy to be 0 when $n \geq 10$, as this typically indicates a pair shares almost identical content, whereas our goal is to find pairs featuring similar topic but different answers. We compute this score for all pairs of documents in $\mathcal{D}$, excluding pairs where both documents are from the same disambiguated entity. If all document pairs receive a score of 0, we disregard $sn$, otherwise, we select one document pair with the highest relevancy score, breaking a tie randomly. This process yields a total of 67,294 document pairs, one per surface name.

## 3.2 Generating Ambiguous Question and Answer Pairs

Having identified two documents with confusing entities that cover a similar topic, we proceed to generate the question $q_{sn}$ that can be answered differently by each document. We use an LLM to generate this: prompting it with $(sn, d_1^*, d_2^*)$ along with two-shot exemplars to generate $\{q_{sn}, [a_1^*, a_2^*]\}$. We initially use GPT-4 (OpenAI, 2023), and then distill its ability to a smaller model, Llama2-13b (Touvron et al., 2023), by finetuning with dataset generated from GPT-4 (West et al., 2021). We obtain a total of 43,839 questions after filtering low-quality examples. Details of the generation and filtering process can be found in Appendix B.2.

## 3.3 Answer Set Expansion

The generated data contains only two answers belonging to two disambiguated entities, while there can be more than two entities sharing the surface name $sn$. In this final stage, we expand the answer set. As we already have a question $q_{sn}$, we find additional answers using an LLM-based QA module. We consider all documents in $\mathcal{D}$, excluding documents that belong to $DE_1^*, DE_2^*$. For each document, we use a Llama2-13b model with 4-shot exemplars to generate an answer given the question (the prompt is in Figure 11 in the appendix). As

| # ans | % Ex. |
|---|---|
| 2 | 49.5 |
| 3 | 26.6 |
| 4 | 13.3 |
| 5 | 6.4 |
| over 5 | 4.3 |

Table 1: Distribution of the number of answers per question (#ans) in AmbigDocs. The maximum number was 10.

Figure 3: Treemap illustrating the distribution of "Wh" questions in AmbigDocs. Each box's size corresponds to its frequency. Questions starting with "What" are the most prevalent, reflecting the dataset's focus on entities.

the QA module can generate an incorrect answer, we keep the newly generated answer only when (1) it passes NLI entailment check (Chen et al., 2021b) and (2) length-normalized log perplexity of the answer is below a threshold value of 0.2. For the NLI entailment check, we consider the document as the *premise*. Additionally, we convert $q_{DE_i^*}$ and $a_i^*$ into a declarative form (Chen et al., 2021b) to use as the *hypothesis*, where $q_{DE_i^*}$ represents a disambiguated question created by replacing $sn$ in the $q_{sn}$ with $DE_i^*$. If multiple answers are found for the same disambiguated entity, we choose the answer with the lowest perplexity.

### 3.4 Dataset Analysis

We collected a total of 36,098 examples, where they are randomly split into train/dev/test splits with ratios of 60%/10%/30%, respectively. We ensure that there is no document overlap between distinct splits. On average, each question has 2.92 answers (distribution in Table 1), covering a total of 102,624 distinct entities. In this work, we do not use training or validation split, only using the test split for evaluation, but will make all data available.

Figure 3 visualizes questions in AmbigDocs. As our generation is synthetic, we (the authors) manually examine 100 instances to assess its quality. We observe two types of noise: when a single document discusses multiple disambiguated entities (2%), and when an LLM generates incorrect answers (4%). Overall, we find question-answer pairs to be mostly valid and answers can be supported by the gold documents. Please refer to Appendix B.3 for further analysis and examples.

## 4 Evaluation for AmbigDocs

We discuss how to evaluate model-generated long-form answer $y$ for question $q_{sn}$ with respect to the reference answer list $[a_1^*, a_2^*, ...a_m^*]$ paired with its disambiguated entity list $[DE_1^*, DE_2^*, \ldots, DE_m^*]$.

### 4.1 Standard QA Metrics

We first report automatic metrics, a token overlap metric and a model-based metric, that has been used in prior work (Adlakha et al., 2023; Stelmakh et al., 2022).

**Token Recall** We map the answer $y$ to a set of tokens by following the answer processing script of SQuADv2 (Rajpurkar et al., 2018) (which lower cases and splits the string) and remove the stopwords from the nltk package. We perform the same preprocessing for individual answer string $a_i^*$ and disambiguated entity name $DE_i^*$ to get token sets. For disambiguated entity name token set, we disregard tokens that appear in the surface name $sn$ to exclude common information and capture the disambiguated entity. For instance, for disambiguated entity name 'Hughesville, New Jersey' in Figure 2, we will only consider {new, jersey}, excluding 'hughesville' which appears in the surface name. Then, we define:

- **Answer Recall**: The average of recall scores of each gold answer $a_i^*$, i.e. $\frac{1}{m} \sum R(a_i^*, y)$, where $R(a_i^*, y)$ represents the token-level recall of $a_i^*$ to $y$.
- **Entity Recall**: The average of recall scores for each disambiguated entities $DE_i^*$. i.e. $\frac{1}{m} \sum R(DE_i^*, y)$. where $R(DE_i^*, y)$ represents the token-level recall of $DE_i^*$ to $y$.
- **Entity-Answer Recall (EAR)**: We introduce a new metric, which averages the product of answer recall and entity recall: $\frac{1}{m} \sum R(DE_i^*, y) \cdot R(a_i, y)$. By multiplying two scores, we measure how many answers are generated with its disambiguated entity name.[5]

**Disambig-F1 (DF1)** Stelmakh et al. (2022) introduces a model-based metric that use a RoBERTa-based (Liu et al., 2019) QA model trained on SQuADv2 to evaluate long-form answer. For each $a_i^*$, the model takes a disambiguated question and the generated answer, and extracts a short answer $a_i$. The F1-score between extracted answer and gold answer is then averaged across all $a_i^*$, i.e. $DF1(y) = \frac{1}{m} \sum F1(a_i, a_i^*)$. To simulate a disambiguated question without explicit human annotation, we replace $sn$ inside $q_{sn}$ with $DE_i^*$ for each gold answer. This metric serves the same purpose as the EAR metric we introduce above.

## 4.2   Answer Categorization

In the overview (Sec. 2), we have defined five types of answers: complete answer, partial answer, ambiguous answer, merged answer, and no answer. We first present manual categorization and then introduce a heuristic to automatically classify answers.

**Manual Annotation** We generate long-form answers for 100 randomly sampled questions in the test set with five LMs (upcoming Sec. 5.1 provides the details of the answer generation process). Given the question $q_{sn}$, the model-generated long-form answer $y$, and gold answers $[a_1^*, a_2^*, ...a_m^*]$ paired with $[DE_1^*, DE_2^*, \ldots, DE_m^*]$, the annotator classifies $y$ into one of the five categories. We provide the instruction for human evaluation in Appendix C.2. One of the authors annotated the initial data, and another person (who has a background in NLP) annotated a subset of the initial data (125 items, 5 model outputs for 25 questions). The Cohen's Kappa statistic for inter-annotation agreement is 0.85, showing high agreement and the validity of our ontology. We will discuss the distribution of model answer types in the results section (manual annotation distribution can be found in Figure 8 in the appendix).

**Automatic Categorization** We develop heuristics to automatically classify the generated answer $y$ into one of the categories proposed in Section 2. Let $T_s$ be the set of tokens mapped from string $s$. Then, we check whether each reference answer pair $(a_i^*, DE_i^*)$ is:

- **answered with disambiguation**: both $a_i^*$ and $DE_i^*$ are mentioned in $y$, i.e. $R(a_i^*, y) \geq \frac{1}{|T_{a_i^*}|}$ and $R(DE_i^*, y) \geq \frac{1}{|T_{DE_i^*}|}$.
- **answered without disambiguation**: $a_i^*$ is mentioned in $y$ but $DE_i^*$ is not, i.e. $R(a_i^*, y) \geq \frac{2}{|T_{a_i^*}|}$[6] and $R(DE_i^*, y) = 0$.

We introduce a moving threshold for recall based on the length of the target string, such as $\frac{1}{|T_{a_i^*}|}$ and $\frac{1}{|T_{DE_i^*}|}$, as longer tokens are less likely to appear verbatim. Then we count the number of **answers with disambiguation**, denoted as $c_p$, and the number of **answers without disambiguation**, denoted as $c_{np}$, for long-form answer $y$. Then, we can classify $y$ into one of five mutually exclusive categories ($m$ is the number of reference answers):

- **Complete Answer**: $c_p = m$
- **Ambiguous Answer**: $c_p = 0, c_{np} = 1$
- **Merged Answer**: otherwise
- **Partial Answer**: $1 \leq c_p < m, c_{np} = 0$
- **No Answer**: $c_p = c_{np} = 0$

---

[5]There may be cases where an answer exploits the metric design, such as '$DE_1^*$ is $a_2^*$ and $DE_2^*$ is $a_1^*$'. However, we have not encountered such cases during the manual inspection of 600 model outputs.
[6]We increase the numerator to reduce false cases. If $|T_{a_i^*}| \leq 2$, we set $R(a_i^*, y) \geq 0.5$.

We provide the pseudocode for classifying answers in Figure 7 in the appendix. Next, we measure the agreement between automatic metrics and manual annotation.

**Verifying Automatic Categorization** We compare the answer category labeled by a human (one of the authors) and the category assigned by our heuristic, on 500 example outputs. The Cohen's Kappa statistic between the two label sets is 0.83, showing strong agreement, though slightly lower than the human agreement of 0.85. We provide the confusion matrix in Figure 5 in the appendix. Some *Ambiguous* answers are misclassified as *Partial* answers due to the limitations of lexical-based heuristics, such as plural forms (locomotive vs. locomotives) and spelling variations (organisation vs. organization).

## 5 Results

### 5.1 Experimental Setup

**Models** We evaluate open-sourced LMs, Llama2 (7b, 13b) (Touvron et al., 2023), Mistral (7b) (Jiang et al., 2023), and two commercial APIs (GPT 3.5 and GPT-4) (OpenAI, 2023).

**Data** We will evaluate the LMs on the test portion of our data, which contains 7,220 questions. For GPT-4 model, we evaluate on 500 random subsets, owning to its expensive API cost. We pair each question with three types of document sets.

- **Gold Only ($n \leq 5$):** The gold documents gathered during data generation process (Kandpal et al., 2022). We randomly sample five documents if there are more than five.
- **Retrieved Only ($n = 5$):** We retrieve five documents with question $q_{sn}$ as the query. We use GTR (Ni et al., 2021) as our retriever and Wikipedia snapshot from December 20th, 2018 as a retrieval corpus. On average, this contains 1.2 gold documents.
- **Gold + Retrieved ($n = 5$):** We take all gold documents, and add top retrieved documents until we have five documents if there are fewer than five gold documents.

**Prompt & Inference** The documents are randomly shuffled before being fed into LMs. We present the prompt used for inference in Figure 11. The answers are greedy decoded.

### 5.2 Standard QA Metric Results

Table 2 presents the model performance on traditional QA metrics. When provided only the gold documents (GOLD ONLY), Mistral and GPT-4 exhibit the strongest performance, followed by Llama2-7b model. Surprisingly, we find some larger models (Llama2-13b, GPT-3.5) significantly perform worse than smaller models (Llama2-7b, Mistral-7b). One potential hypothesis is that larger models with richer parametric knowledge have a bias towards particular contexts or answers, resulting in incomplete answers. Even when provided only gold documents, none of the models achieve an EAR or DF1 score exceeding 0.5, indicating that existing models struggle to distinguish entities that share a surface name. For comparison, we test LMs on the same sets of questions, but providing only one gold document to the model. As opposed to multi-document setting, LLMs can find the single answer, all achieving 0.70-0.80 EAR and above 0.60 DF1 scores (the full results in Appendix D.1).

When retrieved documents are presented with gold documents (GOLD+RETRIEVED), performance declines, suggesting that non-gold but relevant documents distract the model's reasoning process (Cuconasu et al., 2024). Unsurprisingly, solely providing the retrieved corpus (RETRIEVED ONLY) results in a significant performance drop across all models. We find that Mistral model shows the best performance in entity recall in all settings, contributing to its strong overall performance.

**Difference between Entity-Answer Recall and DF1 metrics** Both EAR and DF1 metrics exhibit similar trends, but open-source LLMs show higher EAR scores compared to GPT family, which shows higher DF1 scores. This disparity may be attributed to differences in the

| | Llama2-7b | Mistral-7b | Llama2-13b | GPT-3.5 | GPT-4 |
|---|---|---|---|---|---|
| GOLD ONLY | | | | | |
| Answer Length ($|y|$) | 53.9 | 55.1 | 32.7 | 17.9 | 39.7 |
| Answer Recall | 0.536 | 0.561 | 0.441 | 0.393 | **0.632** |
| Entity Recall | 0.527 | **0.643** | 0.370 | 0.350 | 0.596 |
| Entity-Answer Recall | 0.372 | 0.465 | 0.244 | 0.234 | **0.473** |
| Disambig-F1 | 0.219 | 0.248 | 0.178 | 0.180 | **0.289** |
| GOLD+RETRIEVED | | | | | |
| Answer Length ($|y|$) | 67.0 | 59.9 | 36.0 | 18.4 | 40.6 |
| Answer Recall | 0.495 | 0.483 | 0.383 | 0.337 | **0.509** |
| Entity Recall | 0.520 | **0.571** | 0.351 | 0.319 | 0.482 |
| Entity-Answer Recall | 0.356 | **0.392** | 0.219 | 0.198 | 0.358 |
| Disambig-F1 | 0.189 | 0.199 | 0.144 | 0.151 | **0.222** |
| RETRIEVED ONLY | | | | | |
| Answer Length ($|y|$) | 61.4 | 58.8 | 33.2 | 18.2 | 37.7 |
| Answer Recall | 0.370 | 0.384 | 0.305 | 0.294 | **0.410** |
| Entity Recall | 0.390 | **0.454** | 0.289 | 0.281 | 0.385 |
| Entity-Answer Recall | 0.242 | **0.294** | 0.166 | 0.167 | 0.265 |
| Disambig-F1 | 0.140 | 0.159 | 0.122 | 0.131 | **0.181** |

Table 2: Experimental results on the test split. Answer length is measured by the number of words split by whitespace. We bold the highest score and underline the second-highest score for each metric in each row. Mistral-7b and GPT-4 models exhibit strong performance compared to other models.

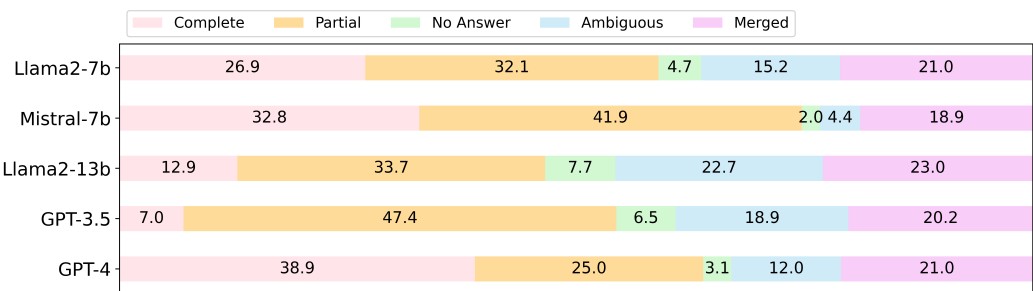

Figure 4: Distribution of answer categories (% in each box) under Gold Only setting. Different LMs show different failure modes.

length of the generated answers. GPT models tend to produce shorter answers compared to open-source LLMs. While lengthy answers may lead to higher recall scores, the QA model may struggle to extract gold answers, resulting in a lower DF1 score.

## 5.3 Answer Categorization Results

We categorize model outputs into five output categories with the automatic metric we have built in Section 4.2. Figure 4 depicts the output distribution under the GOLD ONLY setting for five different models. GPT-4 exhibits strong performance, yielding the highest percentage of complete answer. Mistral generates more *Partial* answers but fewer *Merged* answers, leading to overall performance comparable to GPT-4. Larger models such as Llama2-13b and GPT-3.5 tend to generate more *Ambiguous* and *Merged* answers rather than *Complete* answers, resulting in poor overall performance. A large percentage of GPT-3.5 outputs are labeled as *Partial*. We find GPT-3.5 often generates only one correct answer, as reflected in short length of generated answers. In Table 3, we present an example of generated answers from five LMs.

| **Example** |
| --- |
| $q_{sn}$: When did Senator Grundy serve in the United States Senate? |
| $(DE_1^*, a_1^*)$: (Felix Grundy, 1829 to 1838) |
| $(DE_2^*, a_2^*)$: (Joseph R. Grundy, December 11, 1929, to December 1, 1930) |
| **Predictions** |
| Llama2-7b: (*Merged*) Based on the provided context, Senator Grundy served in the United States Senate from December 11, 1929, to December 1, 1930, and again from October 19, 1829, to July 4, 1838. |
| Mistral-7b: (*Partial*) Felix Grundy served in the United States Senate from October 19, 1829, to July 4, 1838. |
| Llama2-13b: (*Ambiguous*) Based on the provided context, Senator Grundy served in the United States Senate from October 19, 1829, to July 4, 1838. |
| GPT-3.5: (*Ambiguous*) Senator Grundy served in the United States Senate from October 19, 1829, to July 4, 1838. |
| GPT-4: (*Complete*) Joseph R. Grundy served in the United States Senate from December 11, 1929, to December 1, 1930. Felix Grundy served in the United States Senate from October 19, 1829 to July 4, 1838. |

Table 3: Generated answers from five different LMs. The surface name is coloured in purple, where the information relevant to $DE_1^*$ and $DE_2^*$ is coloured in orange and blue, respectively. Classification of each answer is presented next to the model name. LMs exhibit varying behaviours despite given the same prompt.

|  | Llama2-7b | Mistral-7b | Llama2-13b | GPT-3.5 | GPT-4 |
| --- | --- | --- | --- | --- | --- |
| Answer Recall | 0.455 (-.081) | 0.596 (+.035) | 0.466 (+.025) | 0.408 (+.025) | 0.726 (+.094) |
| Entity Recall | 0.444 (-.083) | 0.723 (+.080) | 0.442 (+.072) | 0.397 (+.047) | 0.796 (+.200) |
| Entity-Answer Recall | 0.291 (-.081) | 0.504 (+.039) | 0.288 (+.044) | 0.273 (+.039) | 0.647 (+.174) |
| Disambig-F1 | 0.212 (-.007) | 0.297 (+.049) | 0.215 (+.037) | 0.202 (+.022) | 0.382 (+.093) |

Table 4: Few-shot results under Gold Only setting. We provide the difference from the zero-shot result in the parenthesis.

## 5.4 In-Context Few-shot Learning Results

Prior work shows that in-context examples can help LMs' reasoning process (Liu et al., 2021; Lee et al., 2024). To help LMs, we craft two in-context examples with *complete* answer. Table 4 presents the results under GOLD ONLY setting with two-shot demonstrations (prompt in Figure 12 in appendix). We observe performance gains across all models, except for Llama2-7b, particularly in entity recall. GPT-4 exhibits the most substantial gain among all models, nearly three times more than others. In the appendix (Table 10), we also present the answer category distribution. All models, except for Llama2-7b, show a significant increase in the number of *Complete* answers. We find that larger models generate less *Ambiguous* and *Merged* answers than before, likely due to their learning via few-shot examples. These findings encourage future works on teaching models to distinguish confusing entities and provide *Complete* answers.

## 5.5 Evaluating Answer Precision

So far, we measured the **recall** of the reference answer set. Here, we report K-Precision (Adlakha et al., 2023), the percentage of tokens in $y$ that are from input documents during inference, to evaluate its faithfulness to input documents. We observe the K-precision scores are over 0.84 for all models and evaluation settings, indicating the model generates fairly extractive answers. The full result is reported in Table 11 in the appendix.

# 6    Related Works

**Coreference Resolution**    Our task is closely related to multi-document coreference resolution task (Yang et al., 2015; Cattan et al., 2021) in that it requires entity mention resolution across multiple documents. Yet, our task contains a purposefully ambiguous entity mention in the question. Thus, the entity mention in question can be mapped to multiple entities, making it hard to derive and evaluate against gold entity clusters.

**Complex Reasoning with Retrieval Augmentation**    While language models are equipped with rich parametric knowledge (Roberts et al., 2020), providing relevant documents in context often results in improved performances for various end tasks (Lewis et al., 2020; Asai et al., 2024). Wan et al. (2024); Tan et al. (2024) provides document set containing conflicting answers to contentious questions, while other work (Xie et al., 2023; Chen et al., 2022) simulates a situation where the document set and parametric knowledge of LM collide. WikiHowQA (Bolotova-Baranova et al., 2023) examines multi-document reasoning of LLMs, without engaging conflicting documents. We study a setting where we provide a confusing set of documents which requires careful entity disambiguation. Concurrent to us, Chiang & Lee (2024) studies generating biographies from a set of documents that contains multiple people with the same name. While this work also studies similar entity confusion errors of LMs, its goal is to improve a factuality metric (Min et al., 2023). We cover a much broader set of entities and focus on multi-document reasoning for question answering. Similar to our answer categorization ontology, Mishra et al. (2024) establishes different types of factuality error. Their ontology aims evaluation of general retrieval-augmented generation, while our study focuses on entity disambiguation. Compared to ontology from Chiang & Lee (2024), our ontology is more fine-grained, distinguishing ambiguous and partial outputs.

**Ambiguity in Question Answering**    Prior work studied ambiguous question answering in simple factoid QA (Min et al., 2020; Zhang & Choi, 2021) and long-form QA (Stelmakh et al., 2022) setting. While ambiguity can arise for multiple reasons (Amplayo et al., 2023), we focus on entity disambiguation. Chen et al. (2021a) studied entity linking for the ambiguous entity in a question (e.g., linking ambiguous surface name "Apple" in the question such as "What is the record label of **Apple**?" to its correct Wikipedia entity). Our task setting, which aims to generate a long-form answer covering multiple answers given a question and a set of relevant documents, is equivalent to recent QuoteSum (Schuster et al., 2023) dataset. However, their focus is on generating and evaluating **extractive** answer and does not model whether LMs disambiguate entity mentions. We explicitly simulate confusing document sets and provide metrics to identify whether LMs generate a coherent and complete answer with entity disambiguation.

LLMs can handle ambiguous questions differently than what is explored in this work, which is to present all possible answers with disambiguation (Amplayo et al., 2023; Kim et al., 2023; Sun et al., 2023). These include asking clarifying questions to allow users to specify the intended interpretation of the question (Zhang & Choi, 2023; Lee et al., 2023; Kuhn et al., 2022) or evaluating scores for each candidate answer (Cole et al., 2023).

# 7    Conclusion

This work studies the reasoning capabilities of language models across documents featuring distinct entities that share a surface name. We introduce AmbigDocs, where we identify such documents, paired with an ambiguous question and the answers from each document. Our empirical findings disclose that language models often fail at generating complete answers, exhibiting varying types of behaviours. We hope our findings encourage future investigations on LLMs when faced with entities that are hard to disambiguate.

While we did not explore in this work, the dataset can be used for other tasks such as fact verification (Min et al., 2023) and multi-document summarization (Chiang & Lee, 2024). Future work can also explore fine-grained analysis on the various factors that contribute to determining incomplete answers.

Acknowledgments

We thank the members of UT Austin NLP community for valuable feedback, especially to Hung-Ting Chen, Michael J.Q. Zhang, and Fangyuan Xu. We also thank Hyunji Lee for helpful discussions. The study is partially funded by NSF grant IIS-2312948. This work was supported by Korea Institute for Advancement of Technology (KIAT) grant funded by the Korea Government (Ministry of Education) (P0025681-G02P22450002201-10054408, Semiconductor-Specialized University).

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

## A  Models

Open-source models in our paper are from huggingface. We provide the model names here:

- `Llama-2-7b-chat-hf`
- `Llama-2-13b-chat-hf`
- `Mistral-7B-Instruct-v0.2`
- `question_converter-3b`

- `t5_xxl_true_nli_mixture`
- `gtr-t5-xxl`
- `roberta-base-squad2`

## B  Dataset Generation Details

### B.1  Wikipedia Disambiguation Pages

We use the Wikipedia API to identify disambiguation pages and gather ambiguous entities along with their disambiguations from these pages. We only take that $sn$ have 10 or fewer resolutions, and $DE_i$ that have 20 or fewer documents.

### B.2  Model Distillation

Due to the computational cost associated with GPT-4, we initially generate 1500 questions. Following prior works in automatic dataset construction (Xie et al., 2023; Yehudai et al., 2024), we conduct post hoc filtering to ensure data quality. We divide the filtration into three steps and present the proportion of removed questions during each step in Table 5.

- Not Formatted: We remove the questions if (1) the generated output does not follow the format in Figure 9, or (2) each answer contains too much sentences (answer that contains more than 4 periods).
- Not Ambiguous: We remove the questions if (1) $q_{sn}$ does not contain $sn$, (2) $q_{sn}$ contains both $DE_1^*$ and $DE_2^*$, (3) $q_{sn}$ contains prompt-specific phrases such as 'passage' and 'context', (4) one answer contains another, or (5) two answers are too similar ($\text{IoU}(a_1^*, a_2^*) > 0.75$).
- Not Entailed: We assess the faithfulness of each answer by framing it as a Natural Language Inference (NLI) problem (Bowman et al., 2015). We substitute $sn$ in the question with $DE_i^*$, creating a disambiguated question $q_{DE_i^*}$. We convert $q_{DE_i^*}$ and $a_i^*$ into a declarative form (Chen et al., 2021b) to use as the *hypothesis*, while the gold document serves as the *premise*. We employ T5-11B NLI model (Honovich et al., 2022) to evaluate entailment between *hypothesis* and *premise*, retaining only instances where the *premise* is entailed by *hypothesis* for both answers.

|            | Not formatted | Not ambiguous | Not entail | Success |
|-----------:|:-------------:|:-------------:|:----------:|:-------:|
| GPT-4      | 0.0%          | 8.0%          | 26.5%      | 65.5%   |
| Llama2-13b | 1.7%          | 8.4%          | 23.3%      | 66.6%   |

Table 5: Percentage of questions removed during three filtration steps. Success indicates the questions that were not removed, and therefore collected as our dataset.

This process results in 983 questions, accounting for 65.5% of the original 1500 questions. We distill the generation capabilities of GPT-4 to the Llama2-13b-chat model by finetuning Llama2-13b-chat model with QLoRA (Dettmers et al., 2023) for 5 epochs, with a learning rate of 2e-4. The training script is taken from Yoran et al. (2023). After repeating the same generation and filtration steps to the remaining seed pairs, we obtain a total of 43,839 questions, achieving a success rate of 66.6%, comparable to that of GPT-4.

|                                          | Train  | Dev    | Test   | Total   |
|------------------------------------------|--------|--------|--------|---------|
| Number of questions                      | 25,268 | 3,610  | 7,220  | 36,098  |
| Number of covered disambiguated entities | 71,641 | 10,430 | 20,733 | 102,624 |
| Average number of answers per question   | 2.93   | 2.89   | 2.87   | 2.92    |
| Average answer length                    | 8.48   | 8.58   | 8.38   | 8.47    |

Table 6: Statistics of our dataset. Answer length is measured by the number of words split by whitespace.

| known      | 11.2% | profession | 7.5% | refer | 5.7% | primary  | 5.5% |
|------------|-------|------------|------|-------|------|----------|------|
| population | 5.4%  | located    | 5.0% | main  | 4.8% | location | 4.1% |

Table 7: Frequency of the most common words in the question after being converted to lowercase, removing stopwords, and split by whitespace. The questions fall into two categories: those seeking specific information (profession, population, location, primary, main) and those requesting clarification on ambiguous entities (known, refer).

### B.3 Dataset Analysis

Table 6 presents detailed statistics of our dataset and Table 7 presents the eight most common words appearing in the questions. Additionally, we present three examples of our dataset in Table 8.

Furthermore, we evaluated the quality of 100 instances of our dataset and identified two types of noise. The first occurs when a document discusses multiple entities. For example, the document about the person 'Francisco Prestes Maia' also mentions 'Prestes Maia (building)', which is another disambiguated entity. In such cases, the generated question asks about the building. The second type of noise arises when an answer is incorrect, which typically occurs during the answer expansion process. We leave employing a better QA model and developing better filtration methods for future work.

## C   Evaluation Details

### C.1   Preprocessing

Some surface names contain '(disambiguation)' at the end, which we remove before using $sn$ anywhere. In addition, some disambiguated entities contain phrases enclosed in parenthesis, e.g. 'Michael Jordan (mycologist)'. We remove the parenthesis only when mapping the string into tokens. During mapping, removing token overlaps of $sn$ may result in an empty set of tokens. For such cases, we compute the exact match of the disambiguated entity, i.e. $DE_i^*$ is in $y$.

### C.2   Instruction for Human Annotation

The instructions provided to human annotators include the overview illustrated in Figure 1, the content outlined in Section 2, and a sample annotation interface (Figure 6). For each $y$, the annotator is provided with the follow-

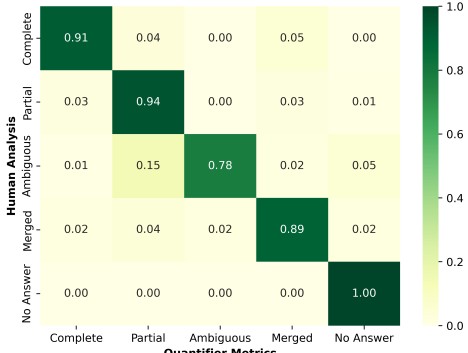

Figure 5: Confusion matrix comparing quantifier metrics with our manual analysis (n=500). Our automatic classification shows high agreement with manual classification, although Ambiguous answers are sometimes misclassified as Partial answers.

---

$q_{sn}$: Who is the artist of the "**Cherry Bomb**" project?

---

$DE_1^*$: Cherry Bomb (EP)
$d_1^*$: Cherry Bomb (EP) Cherry Bomb is the third extended play of **NCT 127**, the Seoul-based sub-unit of the South Korean boy group NCT. It was released by SM ...
$a_1^*$: NCT 127

---

$DE_2^*$: Cherry Bomb (album)
$d_2^*$: Cherry Bomb (album) Cherry Bomb is the third studio album by American rapper **Tyler, the Creator**. It was released on April 13, 2015, ...
$a_2^*$: Tyler, the Creator

---
---

$q_{sn}$: When was **Nengren Temple** first established?

---

$DE_1^*$: Nengren Temple (Guangzhou)
$d_1^*$: Nengren Temple (Guangzhou) Nengren Temple () is a Buddhist temple located ... was first established by Yinjian () in **1824**, during the reign of Daoguang Emperor (1821-1850) of the Qing dynasty (1644-1911). ...
$a_1^*$: 1824

---

$DE_2^*$: Nengren Temple (Jiujiang)
$d_2^*$: Nengren Temple (Jiujiang) Nengren Temple () is a Buddhist temple located ... was first built in the **Northern and Southern dynasties (420-589)**, and went through many changes and repairs through the following dynasties. ...
$a_2^*$: Northern and Southern dynasties (420-589)

---
---

$q_{sn}$: What is the primary use of **Lake Whitney**?

---

$DE_1^*$: Lake Whitney (Connecticut)
$d_1^*$: Lake Whitney (Connecticut) ... Now with a new water treatment facility rated for up to 15 million gallons per day, Lake Whitney has been reconnected as a reserve **water source** for the South Central Connecticut Regional Water Authority. This act has ...
$a_1^*$: Water source

---

$DE_2^*$: Lake Whitney (Texas)
$d_2^*$: Lake Whitney (Texas) Lake Whitney is a **flood control reservoir on the main stem of the Brazos River** in Texas. It is located on River Mile Marker 442 ...
$a_2^*$: Flood control reservoir on the main stem of the Brazos River

---

$DE_3^*$: Lake Whitney Ranch
$d_3^*$: were presented in various town hall meetings where ... Lake Whitney Ranch is a property near Whitney, Texas that is under development as a **summer youth camp, Pathfinder camp, church camp and conference center** for
$a_3^*$: Summer youth camp, Pathfinder camp, church camp and conference center

---
---

Table 8: Three examples from our dataset. The question is presented at the first line, with the bold text indicating the surface name *sn*. Below the question, we present the gold documents, each paired with its disambiguated entity and the answer. Bold text in each document highlights the evidence for the answer.

ing: $(sn, q_{sn}, [DE_1^*, DE_2^*, \ldots, DE_m^*], [a_1^*, a_2^*, ...a_n^*])$. Subsequently, the annotator classifies $y$ into one of the five categories by selecting from a dropdown menu. The distribution of human-annotated answers is presented in Figure 8.

## C.3 Automatic Categorization

Figure 7 describes the pseudocode for classifying a generated answer into one of five categories. We present the confusion matrix comparing automatic metrics and our manual analysis in Figure 5.

| Ambiguous Entity Question | | Disambiguated Entity 1: Gold Answer 1 ... Disambiguated Entity n: Gold Answer n | |
|---|---|---|---|
| | Cluster | Model's Generated Answer | |
| [This is an example] (please feel free to hide) | | | |
| Judge Day What is the state where Judge Day was born? | | Charles Bernard Day: Alabama Edward William Day: Rhode Island William Louis Day: Ohio | |
| | ambiguous ▼ | Judge Day was born in Ohio. | |
| | merged ▼ | Judge Day was born in Alabama, Rhode Island, and Ohio. | |
| | partial ▼ | William Louis Day was born in Ohio, and Charles Bernard Day was born in Alabama. | |
| | complete ▼ | Charles Bernard Day was born in Alabama. Edward William Day was born in Rhode Isl | |
| | fail ▼ | The context does not provide information. | |

Figure 6: Interface for the human annotation process. For each model generated answer, the annoatator is provided the surface name, the question, and the list of reference (disambiguated entity, gold answer) pairs. The dropdown menu contains five categories, where the annotator can easily select one. We do not provide which model $y$ was generated from.

---

**Input:** question $q_{sn}$, reference answer list $[a_1^*, a_2^*, ...a_m^*]$, disambiguated entity list $[DE_1^*, DE_2^*, \ldots, DE_m^*]$, model-generated long-form answer $y$
**Output:** Category $cls$ of $y$ among (Complete answer, Partial answer, No answer, Ambiguous answer, Merged answer)

1: $c_p \leftarrow 0, c_{np} \leftarrow 0$
2: **for** $i \leftarrow 1$ to $m$ **do**
3:     $T_{a_i^*} \leftarrow$ **GetTokens**$(a_i^*)$
4:     $T_{DE_i^*} \leftarrow$ **GetTokens**$(DE_i^*)$
5:     **if** $R(a_i^*, y) \geq \frac{1}{|T_{a_i^*}|}$ **and** $R(DE_i^*, y) \geq \frac{1}{|T_{DE_i^*}|}$ **then**
6:         $c_p \leftarrow c_p + 1$
7:     **else if** $R(a_i^*, y) \geq \frac{2}{|T_{a_i^*}|}$ **and** $R(DE_i^*, y) = 0$ **then**
8:         $c_{np} \leftarrow c_{np} + 1$
9:     **end if**
10: **end for**
11:
12: **if** $c_p = 0$ **and** $c_{np} = 0$ **then**
13:     $cls \leftarrow$ No answer
14: **else if** $c_p = m$ **then**
15:     $cls \leftarrow$ Complete answer
16: **else if** $c_p = 0$ **and** $c_{np} = 1$ **then**
17:     $cls \leftarrow$ Ambiguous answer
18: **else if** $c_p \geq 1$ **and** $c_{np} = 0$ **then**
19:     $cls \leftarrow$ Partial answer
20: **else**
21:     $cls \leftarrow$ Merged answer
22: **end if**
23:
24: **return** $cls$

Figure 7: Algorithm for automatically classifying the generated answer $y$ into one of five categories.

## D   Experimental Details

### D.1   Single Document Study

We provide only one gold document to the LM and examine if the model has identified the correct answer. Since the model is not given multiple entities, the *Ambiguous* answer is resolvable. Therefore, we compute entity recall twice, once with $DE$ and once with $sn$, and select the higher value. We do the same for the DF1 metric, both with and without

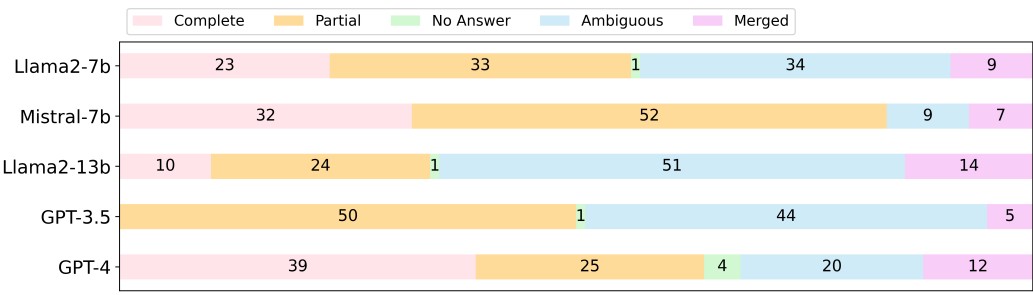

Figure 8: Distribution of human annotated answer categories (% in each box) under Gold Only setting.

|  | Llama2-7b | Mistral-7b | Llama2-13b | GPT-3.5 | GPT-4 |
|---|---|---|---|---|---|
| Answer Recall | 0.784 | 0.852 | 0.768 | 0.804 | 0.855 |
| Entity Recall | 0.985 | 0.988 | 0.990 | 0.943 | 0.956 |
| Entity-Answer Recall | 0.770 | 0.841 | 0.762 | 0.764 | 0.816 |
| Disambig-F1 | 0.644 | 0.613 | 0.656 | 0.625 | 0.648 |

Table 9: Results for the single document study. Results indicate that when provided with only one gold document, the model finds the answer correctly.

replacing $sn$ with $DE$. Table 9 presents the results. Unsurprisingly, all models exhibit the highest answer recall scores throughout our study, as well as other metrics.

### D.2 In-Context Few-shot Learning

Table 10 shows the answer category distribution for few-shot experiment in Section 5.4. All models, except for Llama2-7b, demonstrate improvements in generating *Complete* answers. Moreover, larger models generate fewer *Ambiguous* and *Merged* answers, suggesting that they may have learned how to generate *Complete* answers via few-shot examples.

### D.3 Answer Precision

Table 11 presents K-Precision scores of the models across experimental settings. We observe that the generated answers are well factually grounded to the reference documents.

## E Prompts

We present the prompts used throughout our paper in Figure 9-12.

| | Llama2-7b | Mistral-7b | Llama2-13b | GPT-3.5 | GPT-4 |
|---|---|---|---|---|---|
| Complete | 0.203 (-.066) | 0.435 (+.107) | 0.215 (+.086) | 0.101 (+.031) | 0.586 (+.197) |
| Partial | 0.331 (+.010) | 0.326 (-.093) | 0.303 (-.034) | 0.486 (+.012) | 0.240 (-.010) |
| No Answer | 0.072 (+.025) | 0.017 (-.003) | 0.061 (-.016) | 0.076 (+.011) | 0.012 (-.019) |
| Ambiguous | 0.178 (+.026) | 0.033 (-.011) | 0.193 (-.034) | 0.150 (-.039) | 0.034 (-.086) |
| Merged | 0.217 (+.007) | 0.189 (+.000) | 0.227 (-.003) | 0.186 (-.016) | 0.128 (-.082) |

Table 10: Automatic quantifier metrics for few-shot experiments under Gold Only setting. We provide the difference from zero-shot result in the parenthesis.

| | Llama2-7b | Mistral-7b | Llama2-13b | GPT-3.5 | GPT-4 |
|---|---|---|---|---|---|
| GOLD ONLY | 0.841 | 0.843 | 0.852 | 0.933 | 0.898 |
| GOLD+RETRIEVED | 0.841 | 0.886 | 0.868 | 0.945 | 0.919 |
| RETRIEVED ONLY | 0.850 | 0.898 | 0.875 | 0.949 | 0.934 |

Table 11: K-precision scores on the test split.

You are given two passages about an ambiguous entity with different interpretations. Create a question about the entity, which each passage must answer differently. Find the shortest span of each passage as an answer to each passage.

**Entity**: Young Conservatives
**Passage 1**: Young Conservatives (Czech Republic) | The Young Conservatives () is a political youth organisation in the Czech Republic. It is the youth wing of the Civic Democratic Party (ODS), a centre-right political party, and shares that party's conservative and economically liberal ideology. Young people within the age from 15 to 35 apply for a membership in the MK. Several significant politicians from the ODS party started as members of Young Conservatives, including Jan Zahradil, Jiǒ159ǒ0ed Pospǒ0edǒ161il, Petr Sokol, Martin Baxa, Petr Gandaloviǒ010d, Ivan Langer, Martin Novotnǒ0fd, and Pavel Drobil. Former Chairman of Young Conservatives Petr Mach went on to found a
**Passage 2**: Young Conservatives (UK) | The Young Conservatives (YC) is the youth wing of the Conservative Party in the United Kingdom for members aged 25 and under. The organisation shares the same values and policies as its parent political party with branches being an integrated part of local associations, with the exception of college and university branches which are run independently. YC is both social and political, aiming to bring together young conservatives and encouraging young people to get involved in campaigning. The Ju̇nior Imperial and Constitutional Leaguëwas formed in 1906 with objectives to encourage practical political work and organisation among
**Question**: What is the age range for membership in the Young Conservatives?
**Answer to Article 1**: 15 to 35
**Answer to Article 2**: 25 and under

**Entity**: $\{sn\}$
**Passage 1**: $\{DE_1^*\}$ | $\{d_1^*\}$
**Passage 2**: $\{DE_2^*\}$ | $\{d_2^*\}$
**Question**:

Figure 9: Prompt template for generating question and two corresponding answers with following placeholders: $sn$ for the shared surface name, $d_1^*$ and $d_2^*$ for selected seed documents, and $DE_1^*$ and $DE_2^*$ for its corresponding entities. Due to the limited space, we provide only one-shot example. We expect the model to generate $\{q_{sn}, [a_1^*, a_2^*]\}$, adhering to the format in the template.

Given a passage related to the question, answer to the question. Find the shortest span of the passage as an answer.

**Passage**: Proletarian Unity Party (France) | Proletarian Unity Party (France) The Party of Proletarian Unity (, P̈UP̈) was a French socialist political party. It was formed on December 21, 1930 by leftists expelled from the French Communist Party (PCF), together with some who had previously belonged to the left-wing of the Section franǒ0e7aise de l'Internationale ouvriǒ0e8re (SFIO). Its members were known in France as p̈upistes,̇ and one of its notable leaders was Alexandre Bachelet. Owing to proportional representation, it at one time had ten seats in the Chamber of Deputies of the Third Republic. The PUP affiliated to the London Bureau of left-socialist parties. On January
**Question**: When was the Proletarian Unity Party formed?
**Answer**: December 21, 1930

**Passage**: $\{DE_i\}$ | $\{d_i\}$

> **Question**: $\{q_{sn}\}$
> **Answer**:

Figure 10: Prompt template for answer set expansion with following placeholders: $q_{sn}$ for the question and $d_i$ for the document related to $DE_i$. Due to the limited space, we provide only one-shot example.

> Answer for the given question using only the provided context.
>
> **Question**: $qsn$
> **Context**: $\{DE^*_{f(1)}\} \mid \{d^*_{f(1)}\}$
>
> $\vdots$
>
> **Context**: $\{DE^*_{f(m)}\} \mid \{d^*_{f(m)}\}$
> **Answer**:

Figure 11: Prompt template for inference. We random shuffle the documents and map $d^*_i$ to $d^*_{f(i)}$.

> Answer for the given question using only the provided context.
>
> **Question**: What is the location of the Griswold House?
> **Context**: Griswold House (Guilford, Connecticut) — a limited number of days each week. The Griswold House is located east of Guilford Center, on the south side of Boston Street (Connecticut Route 146) at its junction with Lovers Lane. It is a 2-1/2 story wood frame structure, with a gabled roof, large central chimney, and clapboarded exterior. A leanto section to the rear gives the house a classic New England saltbox appearance. The main facade is three bays wide, with sash windows arranged symmetrically around the entrance. The entrance is flanked by pilasters and topped by a fully pedimented gable. The house was probably built around 1764
> **Context**: John N. A. Griswold House — John N. A. Griswold House The John N. A. Griswold House is a National Historic Landmark at 76 Bellevue Avenue in Newport, Rhode Island. It is home to the Newport Art Museum and houses an art gallery. The home was built in 1864 by Richard Morris Hunt for John Griswold, an Old China Trade merchant and member of the Griswold Family. The house is one of the earliest American Stickž2013style buildings and one of Hunt's first works in Newport. The house is an official project of Americaž2019s Treasures. The Griswold House is a 2-1/2 story wood frame structure, set
> **Answer**: There are two Griswold Houses mentioned in the passages. John N. A. Griswold House is located at 76 Bellevue Avenue in Newport, Rhode Island and Griswold House (Guilford, Connecticut) is located east of Guilford Center, on the south side of Boston Street (Connecticut Route 146) at its junction with Lovers Lane.
>
> **Question**: $q_{sn}$
> **Context**: $\{DE^*_{f(1)}\} \mid \{d^*_{f(1)}\}$
>
> $\vdots$
>
> **Context**: $\{DE^*_{f(m)}\} \mid \{d^*_{f(m)}\}$
> **Answer**:

Figure 12: Prompt template for few-shot learning. We random shuffle the documents and map $d^*_i$ to $d^*_{f(i)}$. Due to the limited space, we provide only one-shot example.

