# OpenReview forum: "AmbigDocs: Reasoning across Documents on Different Entities under the Same Name"
_colmweb.org/COLM/2024/Conference — COLM_

### Official Review · Reviewer_v8yv · 2024-05-02

**Rating:** 9
**Confidence:** 4
**Ethics Flag:** 1

**Summary:**

The paper focuses on evaluation of LLMs in factoid question answering where the salient entities in the question are ambiguous and the answer to the question could change depending on the ambiguity resolution of the salient entities. The authors create AmbigDocs dataset to evaluate multiple LLMs and conclude that LLMs are not currently created with proper strength considering such questions.

**Questions To Authors:**

Suggested write-up improvements:

* Page 1 / Section 1: I'd suggest a retouch on this sentence: " Then, we generate a question and answers from the set of documents, ensuring full annotation of gold document for each answer."

* Page 2/ Section 1: For the sentence: "We find that both open-source LLMs and GPT models suffer at identifying different entities, failing to generate complete answers." I would replace GPT with "OpenAI's GPT" for clarification and I would add a citation for it as well (You have citations for LLaMA and Mixtral earlier but (OpenAI, 2023) is missing and is cited in Section 3.2).

* Page 2/ Section 2 / Task definition: I believe this definition could be improved by citing and properly explaning the task of Retrieval-augmented generation (RAG; Lewis et al., 2020b; Izacard and Grave, 2021; Singh et al., 2021) which is briefly mentioned in the related works section.

* Page 7/ Section 5.1 and 5.2/ Gold only setting: authors may find the work of Kandpal et al. (2022; https://arxiv.org/pdf/2211.08411) related to oracle or gold retrieval settings (for completion of related works coverage).

* Page 8/ Section 5.3: "We categorize model outputs into five output categories with the automatic metric we have
built in Section 4.2" a period is missing at the end of this sentence.

**Reasons To Accept:**

* The paper studies the knowledgabilty of LLMs and the factuality of their generated answers in a realistic setting.
* The idea is quite well thought and the paper asks the right questions and properly answers those questions.
* Multiple LLMs are studied so the results are more trustable.

**Reasons To Reject:**

* The write up has a few missing citations and could be improved.

Please refer to the suggested improvements in the next section.

---

> ### Author Rebuttal · Authors · 2024-05-30
>
> Thank you for your review and we are excited to see that you found our paper to present a realistic idea. Thank you for the detailed suggestions, we will make edits incorporating your suggestions.

---

> > ### Comment · Reviewer_v8yv · 2024-06-05
> > **Rebuttal**
> >
> > Rebuttal acknowledged.

---

### Official Review · Reviewer_dFmw · 2024-05-08

**Rating:** 8
**Confidence:** 4
**Ethics Flag:** 1

**Summary:**

Distinguishing between different entities that share the same name poses a significant challenge for language models (LMs). The paper addresses this challenge by introducing a novel benchmark, AmbigDocs, which leverages Wikipedia's disambiguation pages to identify documents related to entities sharing an ambiguous name. The method involves generating questions containing ambiguous names and their corresponding sets of answers from these documents. The goal of this benchmark is to evaluate the ability of LMs to reason across multiple documents with ambiguous entities and provide cohesive answers despite the confusion caused by entities with the same name.

**Questions To Authors:**

1. Regarding seed document pair generation (Section 3.1), could the authors elaborate on how the set of documents D is collected from Wikipedia's disambiguation pages? Are there any specific criteria used to select these documents? How to ensure the quality and relevance of the documents obtained from the disambiguation pages?

2. Also in Section 3.1, the method for calculating the relevancy score between document pairs based on n-gram overlap is interesting. However, could the authors explain why a threshold of 3 and 10 is chosen for determining relevancy? How was this threshold decided? The paper mentioned setting relevancy to 0 when n ≥ 10 to indicate almost identical content. However, could this threshold potentially exclude relevant document pairs where the content is similar but not identical?

3. In section 3.2, the paper mentioned obtaining a total of 43,839 questions after filtering low-quality examples. What criteria were used to determine the quality of the generated questions? Could the authors provide some examples of what constitutes a low-quality question in this context?

4. In section 3.3, the paper mentioned two criteria for retaining newly generated answers: passing an NLI entailment check and having a length-normalized log perplexity below a threshold value of 0.2. Could the authors explain the rationale behind these criteria? How effective are these criteria in filtering out incorrect or irrelevant answers?

**Reasons To Accept:**

1. The paper introduces a new benchmark, AmbigDocs, designed specifically to tackle the task of reasoning across documents with entities that share the same name. This benchmark provides a solid resource for evaluating the performance of LMs in handling such ambiguity.

2. Empirical findings presented in the paper shed light on the performance of current LLMs in generating answers for questions involving ambiguous entity mentions. By categorizing incomplete answers and establishing an ontology for evaluation, the paper offers insightful analysis into the challenges faced by LMs when dealing with ambiguity.

**Reasons To Reject:**

1. The paper proposes an ontology categorizing incomplete answers and automatic evaluation metrics to identify them. However, how accurate are these metrics in distinguishing between incomplete answers caused by genuine ambiguity and those due to other factors such as model limitations or dataset biases?

2. In the section of "3. Creating AmbigDocs", some details still require further clarification. Please see my questions in "Questions To Authors".

---

> ### Author Rebuttal · Authors · 2024-05-30
>
> Thank you for your review and we are delighted to see that you found our paper to offer insightful analysis. We address your concerns below:
>
> [R1] Distinguishing between genuine ambiguity and other factors
>
> We assume that LMs should ideally generate complete answers as all necessary information is provided in-context, thereby not considering external factors such as dataset bias during evaluation. Future work could perform fine-grained analysis on the various factors that contribute to determining incomplete answers.
>
> [Q1] Collection of documents
>
> We use the Wikipedia snapshot from December 20th, 2018, where each article is split into 100 words (which we consider as documents) [1]. For each disambiguated entity of the snapshot, we collect the documents where the disambiguated entity serves as the title.
>
> [Q2] Threshold for calculating the relevancy score
>
> Yes, we agree we might be missing some potential document pairs. We focused more on precision of identified document pairs, to manage the computational load of data generation in Section 3.2. We ran some pilot studies to balance the exploration of all possible document pairs against limited computational budget and manually selected threshold value. We found that documents with a maximum 3-gram overlap were mostly independent, whereas those with high n-gram overlap often contained the exact same sentences.
>
> [Q3] Filtering low-quality examples
>
> We removed examples that were not formatted, were not ambiguous, and failed to pass the NLI check. For details, please refer to Appendix B.2.
>
> [Q4] Retaining newly generated answers
>
> We follow prior works to evaluate the QA model’s correctness by converting it into a NLI problem [2, 3]. When multiple answers were found, we computed the perplexity of each answer to select the most adequate one, where we also observed that answers with perplexity above 0.2 were often wrong. Following [3], we manually evaluated 100 instances, where 4% of questions included a single incorrect answer in their set of answers (Appendix B.3). We will make this clearer in the revision.
>
> [1] Karpukhin, Vladimir, et al. Dense Passage Retrieval for Open-Domain Question Answering. EMNLP 2020
>
> [2] Chen, Jifan, et al. Can NLI Models Verify QA Systems’ Predictions? EMNLP 2021 Findings
>
> [3] Xie, Jian, et al. Adaptive chameleon or stubborn sloth: Revealing the behavior of large language models in knowledge conflicts. ICLR 2024

---

> > ### Comment · Reviewer_dFmw · 2024-06-06
> >
> > Rebuttal acknowledged.

---

### Official Review · Reviewer_mQPG · 2024-05-12

**Rating:** 7
**Confidence:** 3
**Ethics Flag:** 1

**Summary:**

The paper defines a methodology for analysing entity disambiguation performance of LLMs. A part of the methodology is the creation of a large scale data set that leverages Wikipedia for entity disambiguation.

I have concerns about this paper because it misses a baseline. There are many Wikipedia based tools that allow entity disambiguation, e.g.
https://www.dbpedia-spotlight.org/

In addition also since several years, in industry NLP libraries there are tools to create entity linking based on custom data, e.g.

https://github.com/explosion/projects/blob/master/nel-emerson/scripts/notebook_video.ipynb

Without a baseline using existing tools or an existing language model data set (e.g. based on Spacy), the usefulness of the new benchmark is hard to evalute.

**Reasons To Accept:**

A novel benchmark about entity disambiguation for LLMs is introduced.

**Reasons To Reject:**

The benchmark misses a baseline against which the LLMs are compared. I am withdrawing this concern, the authors explained in the rebuttal that they have something else in mind.

---

> ### Author Rebuttal · Authors · 2024-05-30
>
> Thank you for your review and we are glad to see that you found our benchmark to be novel. We address your concerns below:
>
> [R1] Comparing with baselines using existing Entity Linking tools
>
> This work evaluates LM’s long-form answer generation ability, while distinguishing confusing entities. Entity linking, on the other hand, links existing entity mention in the input text to disambiguated Wikipedia pages. Because of this major difference, it is not clear how to use entity disambiguation tools for this task. We will add discussions on this in the final draft.

---

> > ### Comment · Reviewer_mQPG · 2024-06-05
> > **Rebuttal**
> >
> > Rebuttal acknowledged.

---

### Official Review · Reviewer_JT3K · 2024-05-15

**Rating:** 6
**Confidence:** 4
**Ethics Flag:** 1

**Summary:**

This paper proposes AmbigDocs, a new benchmark for studying the behavior of LLMs on ambiguous document-based QA, and a methodological pipeline that can be used to create similar benchmarks from Wikipedia disambiguation pages using LLMs. The questions in the benchmark are ambiguous in the sense that they contain a surface form of a named entity, which may refer to multiple actual entities in the answer documents.

The paper also gives a new spin to the definition of an answer in the context of LLMs and document-based QA and provides a taxonomy of 5 answer types, against which several SOTA closed- and open-source LLMs (LLaMa 7b and 13b, Mistral 7b, GPT-3.5 and GPT-4) are evaluated. The benchmark and the answer taxonomy are the main contributions.

The key findings are twofold: 1) generating complete answers for ambiguous questions in AmbigDocs is a challenging task for the open- and close-source LLMs 2) prompting with in-domain in-context examples significantly improves the performance of LLMs.

The paper is generally well-written and the benchmark creation pipeline is clearly presented, although the analysis of results is quite short and shallow, which is the main limitation of this paper. Another weakness is the artificial nature of ambiguity in benchmark questions and its unclear practical applications.

**Questions To Authors:**

1. Some of your findings are quite counter-intuitive (e.g. LLaMa 2 with 13b parameters performing worse than the one with 7b parameters, Mistral demonstrating the best overall performance). Do you have any intuition why?
2. What are some practical applications of your benchmark besides examining the performance of LLMs on the ambiguous QA task?

**Reasons To Accept:**

1. The behavior of LLMs on ambiguous document-based QA has not been studied before
2. Taxonomy of answer types with heuristics for answer classification

**Reasons To Reject:**

1. Detailed analysis of results is lacking
2. The ambiguity in questions seems quite artificial, unclear whether this benchmark has any practical applications

---

> ### Author Rebuttal · Authors · 2024-05-30
>
> Thank you for your review and we are happy to see that you found our paper to investigate behaviors of LLMs that have not been studied before. We address your concerns below:
>
> [R1] Detailed analysis of results
>
> We will strengthen our analysis in the final version of our paper. Examining the performance across different types of questions (e.g. Figure 3, Table 6) could enable a more fine-grained understanding of the results. Manually coding model prediction comparison can provide deeper understanding of model performance as well (e.g. example in appendix Table 8).
>
> [R2] Artificial ambiguity of questions
>
> This is a valid concern as our dataset is synthetically generated. Upon manual inspection, however, we find that our questions are mostly reasonable. Please see Section 3.4 and Appendix B.3 for our random samples of examples.
> We also emphasize that entity ambiguity – “noun phrase in the question which may be resolved in multiple ways” [2], the problem we study, occurs naturally in human interaction data [1].
>
> [Q1] Intuition behind counter-intuitive findings
>
> Great question! As these models are released without precise description on how they are trained, it is a bit hard to understand why. One hypothesis is that larger models, with richer parametric knowledge, have a bias towards particular contexts or answers, resulting in incomplete answers.
>
> [Q2] Applications besides ambiguous QA
>
> Our benchmark offers challenging document sets for tasks involving multi-document reasoning, which can be used to evaluate multi-document summarization and fact verification. I.e., Given the confusing document set, see whether LLM can generate a coherent summary. We will discuss alternative use cases of our datasets in the revision.
>
> [1] Min, Sewon, et al. AmbigQA: Answering Ambiguous Open-domain Questions. EMNLP 2020
>
> [2] Amplayo, Reinald Kim, et al. Query Refinement Prompts for Closed-Book Long-Form QA. ACL 2023

---

> > ### Comment · Reviewer_JT3K · 2024-06-05
> > **Re: Rebuttal by Authors**
> >
> > I read the authors' responses and chose to keep my original evaluation.

---

### Comment · Area_Chair_LzXR · 2024-06-02
**Discussion period now open**

Hi reviewers, please take a look at the author's rebuttals and the other reviews for this paper!

If the rebuttals addressed your concerns, please let the authors know about this and update your review. If not, please continue to engage with the authors and the other reviewers in the discussion forum.

It seems there is significant disagreement among the reviewers about this paper.

mQPG, you are the most negative about the paper, and your concern is that the paper doesn't compare with existing benchmarks for the task. Does the authors' response clarify this concern, or do you still hold this concern?

dFmw and v8yv, you are both very positive about the paper. Would you like to argue for its acceptance?

---

> ### Comment · Reviewer_dFmw · 2024-06-06
>
> Hi AC and all,
>
> After reviewing all the comments and rebuttals, I still retain my score (according to ACL/EMNLP standards). I share reviewer JT3K's concerns regarding the analysis of experimental results, and I also found some details in methodology unclear (although the author's rebuttal has addressed my questions). However, I believe these are minor issues; differing views and debates indicate the community's interest in this paper. Consider that this paper is the first to study the performance of LLMs on ambiguous entities (the same problem we've also observed affecting retrievers in RAG-QA, despite mitigation strategies in GPT-4 and Claude 3), introducing this perspective is significant. If this paper is accepted, I suggest that AC remind the authors in the meta-review to address the reviewers' concerns as much as possible in the camera-ready version.
>
> Many thanks. --dFmw

---

### Decision · Program_Chairs · 2024-07-10

**Decision:**

Accept

**Comment:**

This paper studies the problem of ambiguous entity names, e.g., where two well-known people share the same name and there are a variety of documents that refer to each of the different entities. The paper creates a new dataset exemplifying this challenge, including questions that require disambiguating ambiguous entity names (e.g., a name two people share) to a specific individual entity. Authors perform analysis on the performance of SOTA models and categorize their failures to disambiguate entities in to several error categories. This problem is particularly important for applications of such SOTA systems, e.g., for RAG tasks.

One concern that reviewers hold is that questions are synthetically generated and may not reflect actual naturalistic scenarios where entity disambiguation is required. However, reviewers are generally positive about the creation of this benchmark as a way to directly investigate LLM performance on this important problem.

If accepted, the authors should ensure that they address reviewer concerns and suggestions in the final version of the paper.